# $\mu_3$-Oxo nucleophile formation enables efficient $S_N2$ hydrolysis at the trinuclear metal center in inorganic pyrophosphatase

Saki Maruoka[1,2], Yohei Kametani [3], Eisuke Magome [4], Hiroyuki Setoyama[4], Masahide Kawamoto [4], Masaki Horitani [2], Takamasa Teramoto [1], Yoshimitsu Kakuta [1], Yoshihito Shiota [3], Kazunari Yoshizawa [3,5] & Keiichi Watanabe [2,6] ✉

Inorganic pyrophosphatases are essential metalloenzymes for phosphate metabolism. Bacterial Family II Inorganic pyrophosphatases utilize a trinuclear metal center and exhibit higher catalytic activity than binuclear counterparts. Here we show the mechanism underlying this enhanced hydrolytic efficiency in the enzyme from *Shewanella* sp. AS-11 using X-ray absorption spectroscopy, site-directed mutagenesis, and density functional theory calculations. We identify a catalytic $\mu_3$-oxo nucleophile—generated by proton transfer from a bridging $\mu_3$-hydroxide to Asp14—as the key reactive species for hydrolysis. Rotation of Asp14 drives this conversion and constitutes the rate-limiting step, with an activation barrier of 15.5 kcal mol$^{-1}$. The trinuclear metal center promotes hydrolysis by lowering the p$K_a$ of the hydroxide to facilitate $\mu_3$-oxo formation, stabilizing this intermediate, positioning the nucleophile for optimal in-line attack, and enhancing phosphorus electrophilicity. These findings highlight the importance of reactive species generation and illustrate how metalloenzymes exploit geometric and electronic tuning to achieve high catalytic reactivity.

Nature employs highly sophisticated strategies to precisely position metal ions within enzyme active sites, thereby enabling accurate and efficient hydrolytic reactions. Many hydrolases utilize two divalent metal cations, such as $Zn^{2+}$, $Mg^{2+}$, $Ni^{2+}$, $Mn^{2+}$, or $Fe^{2+}$, to coordinate a water-derived oxygen that functions as a nucleophile[1–3]. These structural adaptations underlie the remarkable specificity and catalytic efficiency observed in biological systems, supporting essential functions such as metabolism and regulation.

A prominent example is the cleavage of phosphoanhydride and phosphoester (P–O) bonds in phosphate metabolism, particularly in pathways involving inorganic pyrophosphate (POP) and its derivatives[4,5]. Hydrolytic reactions of this type are catalyzed by phosphatases and pyrophosphatases, which are primarily classified by the nature of their nucleophiles and metal coordination modes[6]. Representative nucleophilic species include cysteine thiolates in tyrosine phosphatases (type A)[7,8], serine alkoxides in PhoA-type alkaline phosphatases (type B)[9,10], terminal hydroxides coordinated to a single metal ion as in purple acid phosphatases and

3'-5' exonucleases (type C)[11–13], and bridging hydroxides stabilized by two metal ions as seen in phosphotriesterases and serine/threonine phosphatases (type D)[14–16]. A distinct $\mu_3$-oxo species coordinated by three metal cations has been observed in PhoX-type alkaline phosphatases from Pseudomonas fluorescens (type E)[17].

In this study, we focused on inorganic pyrophosphatases (PPases), a class of essential enzymes involved in intracellular phosphate metabolism (Fig. 1a)[18,19]. Previous X-ray crystallographic studies of Family II PPases from *Shewanella* sp. AS-11 (ShPPase) and *Bacillus subtilis* (BsPPase), complexed with a substrate analog imidodiphosphate (PNP), have revealed a unique trinuclear metal center. Each metal ion coordinates both a terminal oxygen of the electrophilic phosphate and a lone pair of the putative nucleophile, aligning them linearly with the scissile bond (Fig. 1b, c)[20,21]. However, it has remained unclear how the nucleophile—a $\mu_3$-hydroxide with all three lone pairs engaged in metal coordination—can attack the electrophilic phosphorus center (Fig. 1b).

[1]Laboratory of Biophysical Chemistry, Department of Bioscience and Biotechnology, Faculty of Agriculture, Kyushu University, Fukuoka, Japan. [2]Department of Applied Biochemistry and Food Science, Saga University, Saga, Japan. [3]Institute for Materials Chemistry and Engineering and IRCCS, Kyushu University, Fukuoka, Japan. [4]SAGA Light Source, Tosu, Japan. [5]Fukui Institute for Fundamental Chemistry, Kyoto University, Kyoto, Japan. [6]Department of Data Science in Food Environment, Kyushu Nutrition Welfare University, Kitakyushu, Japan. ✉e-mail: watakei@knwu.ac.jp

**Fig. 1 | Catalytic function and active site structures of ShPPase. a** Schematic of the pyrophosphatase-catalyzed hydrolysis reaction. **b** Model depicting the proposed nucleophilic attack by a $\mu_3$-OH$^-$ species based on crystallographic data. Black dashed lines indicate coordination bonds; the yellow triangle denotes the plane defined by the trinuclear metal cluster. **c** Coordination geometries of the metal ions in the active sites of the substrate-free Mn$^{2+}$–ShPPase (PDB: 6LL7, left) and the PNP-complexed Mg$^{2+}$–ShPPase (PDB: 6LL8, right). Amino acid residues and the substrate analogue PNP are shown as sticks; metal ions are represented as spheres. A five-coordinate geometry is observed at M1 and M2 in the substrate-free state, while a six-coordinate geometry is seen in the PNP-bound state. Mn, Mg, O, N, P, and H atoms are colored purple, green, red, blue, orange, and white, respectively.

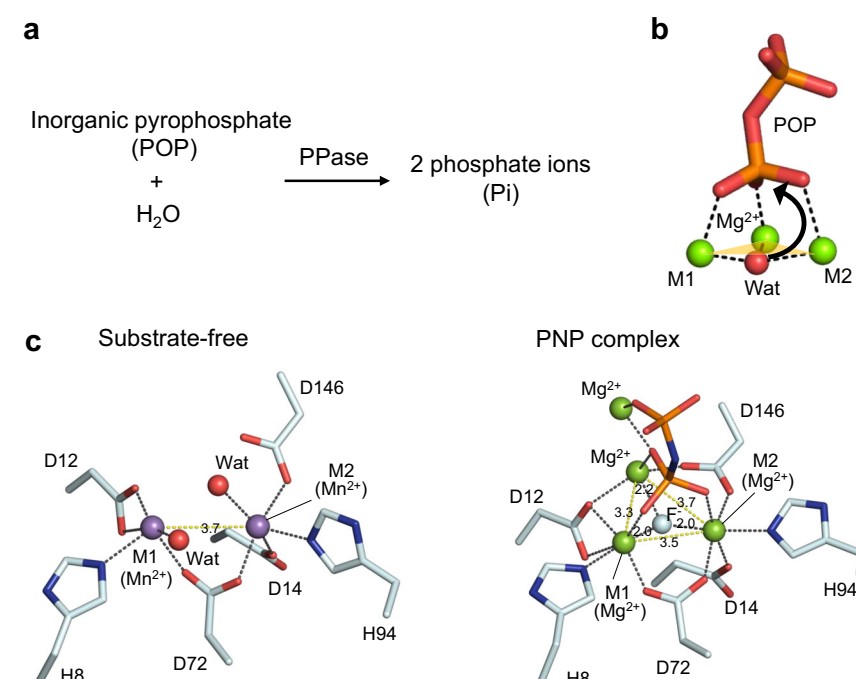

## Table 1 | Kinetic parameters of Zn$^{2+}$-ShPPases

| Zn$^{2+}$-ShPPase | $k_{cat}$ (s$^{-1}$) | $K_m$ (μM) | $k_{cat}/K_m$ (s$^{-1}$ μM$^{-1}$) |
|---|---|---|---|
| Wild type | 99.8 ± 5.5 | 79 ± 18 | 1.26 |
| D14A | 0.019 ± 0.001 | 179 ± 37 | 0.00011 |
| D14N | 0.034 ± 0.002 | 100 ± 20 | 0.00034 |

Kinetic parameters determined from activity assays. Values represent the mean ± s.d. from $n = 3$ independent experiments.

To address this question, we employed X-ray crystal structure-based extended X-ray absorption fine structure (XCS-EXAFS) spectroscopy, site-directed mutagenesis, and density functional theory (DFT) calculations using a trinuclear metal cluster model of the active site. Our mechanistic analysis reveals that, unlike the canonical μ-hydroxide-bridged binuclear mechanism (type D), hydrolysis by ShPPase proceeds via a distinct $\mu_3$-oxo intermediate formed within the trinuclear metal center. Notably, the side chain of the conserved general base Asp14 undergoes a conformational rotation to accept a proton from the $\mu_3$-hydroxide, forming the active $\mu_3$-oxo nucleophile. This mechanistic feature provides a clear explanation for the higher catalytic efficiency of Family II PPases compared to their binuclear Family I counterparts[18,22]. Our findings highlight the importance of reactive species generation in catalytic performance and illustrate how metalloenzymes exploit geometric and electronic tuning to achieve high reactivity, offering valuable insights for the design of biomimetic catalysts and the engineering of metalloenzyme functions.

## Results and discussion
### EXAFS analysis of substrate-free and PNP-bound ShPPase
In the substrate-free state, the active site of Family II ShPPase accommodates two divalent transition metal cations at the M1 and M2 sites[18,20]. Previous crystallographic studies have shown that Mn$^{2+}$ ions occupy these positions in the substrate-free form (Mn$^{2+}$-ShPPase, PDB: 6LL7) (Fig. 1c, left)[20]. In contrast, upon binding of the substrate analog PNP, an additional metal cation (Mg$^{2+}$) is incorporated into the dinuclear center to generate a trinuclear metal site (Mg$^{2+}$-ShPPase, PDB: 6LL8) (Fig. 1c, right)[20]. In this structure, the inhibitor F$^-$ is positioned at the center of the metal cluster, and is presumed to be replaced by a nucleophilic OH$^-$ species during catalysis. To elucidate how a $\mu_3$-hydroxide species, lacking a free lone pair, could

participate in nucleophilic attack on the phosphorus atom, we performed high-resolution EXAFS analysis (with 0.01 Å-level resolution) to determine the metal coordination geometry in solution[23–26].

The metal ions at the M1 and M2 sites of ShPPase are exchangeable in vitro, and previous studies have shown that hydrolytic activity is retained with various divalent transition metals[27]. For EXAFS measurements, we selected Zn$^{2+}$-activated ShPPase (Zn$^{2+}$-ShPPase), which exhibits both strong X-ray absorption (Supplementary Fig. 1) and sufficient enzymatic activity (Table 1). Comparative Zn K-edge XAFS spectra of the substrate-free and PNP-bound forms reveal distinct differences in both the XANES and EXAFS regions (Fig. 2). In the XANES region, the white line—an intense feature immediately above the absorption edge—increases upon PNP binding (Fig. 2a), consistent with the coordination change observed in the crystal structures. The substrate-free enzyme is predominantly five-coordinate, whereas the PNP complex is six-coordinate; the enhanced white-line intensity therefore reflects this coordination shift. In the k$^3$-weighted Fourier transform of the EXAFS (Fig. 2b), the PNP complex exhibits a stronger first-shell peak and sharper outer-shell features, indicating an altered local environment around Zn in the PNP-bound state.

Based on the Mn$^{2+}$-ShPPase crystal structure, the first coordination shell likely corresponds to 4–5 oxygen and nitrogen atoms located approximately 2.5 Å from Zn$^{2+}$ (Figs. 1c and 2b). Outer-shell features arise from single and multiple scattering paths involving histidine and aspartate residues, as well as the phosphorus atom of PNP, consistent with R values in the 3–5 Å range. The relatively high symmetry of Zn–Zn distances observed in the crystal structure allowed identification of four major single-scattering paths: Zn–O, Zn–N, Zn–C, and Zn–Zn (Supplementary Fig. 2a).

Curve fitting in k-space using parameters derived from the crystal structure revealed dynamic changes in the coordination environment upon PNP binding. Prior studies have demonstrated that metal substitution in the active site does not significantly perturb the overall structure[20,21,28], validating our experimental approach. We performed the fitting analysis using constraint parameters as described in the Methods section[23] (Table 2 and Supplementary Fig. 2b). The results showed that Zn$^{2+}$ ions are predominantly five-coordinate in the substrate-free form, but transition to six-coordinate upon PNP binding. This change aligns with the crystallographic data and accurately reflects the altered coordination environment (Table 2 and Fig. 1c). These findings highlight the sensitivity of EXAFS in capturing subtle changes in metal coordination geometry and support the use of this

**Fig. 2 | XAFS analysis of Zn²⁺–ShPPase. a** Zn K-edge XANES spectra of the substrate-free (light blue) and PNP-bound (black) forms. **b** Fourier-transformed EXAFS spectra. Light blue and black lines correspond to experimental data for the substrate-free and PNP-bound forms, respectively.

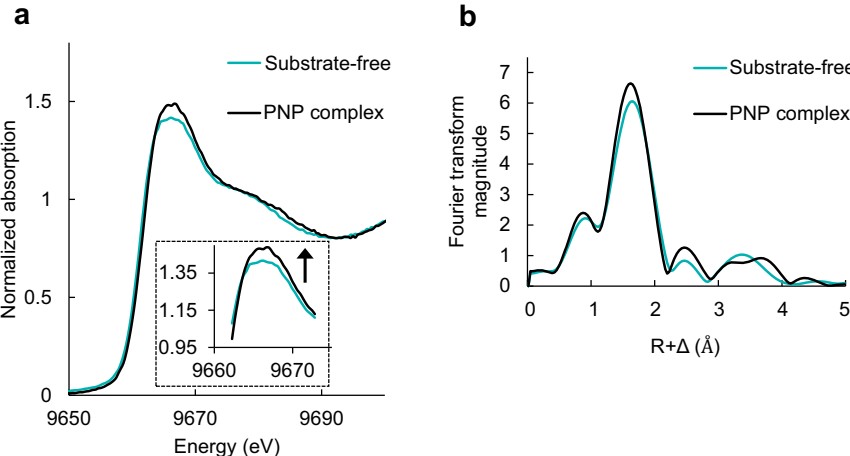

## Table 2 | Best fits for EXAFS data

| | Site | Absorber-Scatterer | $N$ | $R$ (Å) | $\sigma^2$ | $E_0$ | $R_{factor}$ |
|---|---|---|---|---|---|---|---|
| Substrate-free | M1 | Zn-O | 3.5 | 2.06 | 0.0075 | 9.999 | 0.784 |
| | | Zn-N | 1 | 2.56 | | | |
| | | Zn-Zn | 1 | 3.97 | | | |
| | M2 | Zn-O | 3.5 | 2.05 | 0.0069 | 9.998 | 0.871 |
| | | Zn-N | 1 | 2.57 | | | |
| | | Zn-Zn | 1 | 3.94 | | | |
| PNP complex | M1 | Zn-O | 4 | 2.02 | 0.0034 | 3.478 | 0.035 |
| | | Zn-O ($\mu_3$) | 1 | 1.82 | | | |
| | | Zn-N | 1 | 2.14 | | | |
| | | Zn-P | 1 | 3.31 | | | |
| | | Zn-Zn | 1 | 3.33 | | | |
| | M2 | Zn-O | 4 | 2.11 | 0.0126 | 5.684 | 0.029 |
| | | Zn-O ($\mu_3$) | 1 | 1.82 | | | |
| | | Zn-N | 1 | 2.13 | | | |
| | | Zn-P | 1 | 3.27 | | | |
| | | Zn-Zn | 1 | 3.32 | | | |

Each row corresponds to a best fit for data obtained from measurements on an independently prepared set of samples.
$N$ coordination number, $R$ interatomic distance (Å), $\sigma^2$, meansquare deviation in $R$ (the Debye-Waller factor) (Å²), $E_0$, threshold energy shift (eV).

technique for probing metalloenzyme active sites under physiologically relevant conditions.

### Structural evidence for a µ₃-oxo nucleophile in the PNP-bound complex

EXAFS fitting revealed that the Zn–O (water) bond distance is significantly shortened from 2.06 Å in the substrate-free state to 1.82 Å in the PNP-bound complex, indicating a stronger coordination induced by substrate binding (Table 2). The Zn–Zn distance also decreased from 3.96 Å to 3.33 Å, suggesting substantial structural reorganization within the trinuclear metal center. Although subtle, these changes were clearly resolved by EXAFS, demonstrating its sensitivity to fundamental alterations in metal coordination environments at the active site.

A particularly noteworthy feature is the presence of a Zn–O($\mu_3$) bond at 1.82 Å, identified using an independent fitting parameter among the five oxygen atoms coordinating each Zn²⁺ ion. To validate this assignment

given the potential presence of uninuclear species in the sample, we performed systematic fitting tests (Supplementary Table 3). Fixing the Zn–O distance to a typical non-bridging water coordinate (2.0 Å) resulted in a statistically significant increase in the R-factor (from 0.035 to 0.047) and $\chi2$ (from 67.05 to 89.24), confirming that the short shell is indispensable for an accurate fit. Furthermore, because the uninuclear population likely possesses longer Zn–O bonds (2.0 Å), the extracted 1.82 Å value represents an ensemble average and thus serves as a conservative upper limit for the actual bond length in the active multinuclear center. This result supports the assignment of a $\mu_3$-oxo (O²⁻) species[24,25] bridging two Zn²⁺ ions and one Mg²⁺ ion within the active site. The validity of this interpretation is further supported by the Fe–O distance of 1.80–1.83 Å observed for the bridging oxo group in the binuclear iron center of oxyhemerythrin[26].

The formation of a $\mu_3$-oxo species implies deprotonation of a hydroxide ion to generate an oxo species, which plays a critical role in catalysis by facilitating nucleophilic attack on the phosphorus atom of inorganic pyrophosphate (POP). This structural transformation is therefore central to the enzymatic mechanism of phosphate bond cleavage.

### Construction and optimization of a computational model of the ShPPase active site

To further validate the structural insights obtained from EXAFS, we performed DFT-based mechanistic modeling using the high-resolution crystal structure of the PNP-bound complex (PDB ID: 6LL8). Prior to mechanistic analysis, we constructed and evaluated a computational model that included PNP, four metal ions, and a fluoride ion occupying the position corresponding to the nucleophilic water molecule. This model was optimized using DFT, yielding an average Zn–O bond distance of 2.13 Å, which is in good agreement with the 2.0 Å observed in the crystal structure, thus supporting the validity of the model. Based on this foundation, we constructed both a PNP-bound model (representative of the EXAFS experimental condition) and a model for the catalytic reaction with inorganic pyrophosphate (POP), by substituting specific atoms or ligands as detailed in the Methods section and Supplementary Figs. 3–6.

To better reproduce the enzyme environment used for EXAFS measurements, the crystal structure model was modified by replacing fluorine with hydroxide or oxygen and Mg²⁺ at the M1 and M2 sites with Zn²⁺. In these optimized models, the calculated Zn–O ($\mu_3$) bond distances were 2.22 Å for the $\mu_3$-hydroxide state and 1.97 Å for the $\mu_3$-oxo state (Supplementary Figs. 4 and 7). The shorter distance in the $\mu_3$-oxo configuration supports the presence of an oxo ion (O²⁻), consistent with previous DFT studies of other metalloprotein cores[6,17]. Additionally, the Zn–Zn distances decreased from 3.68 Å in the $\mu_3$-hydroxide state to 3.28 Å in the $\mu_3$-oxo state, further supporting a compaction of the metal cluster upon formation of the oxo species. These computational results align closely with our EXAFS data

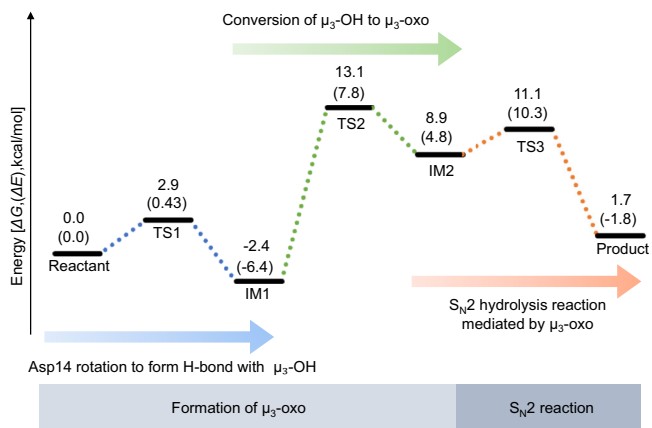

**Fig. 3 | Free energy profile for the POP hydrolysis reaction.** Gibbs free energy changes (ΔG, kcal mol⁻¹) are shown as the primary values, with the corresponding electronic energy changes (ΔE) given in parentheses. The reaction pathway consists of Asp14 rotation to form H-bond with μ₃-OH (blue dashed line), Conversion of μ₃-OH to μ₃-oxo (green dashed line), and S$_N$2 hydrolysis (orange dashed line).

and strongly support the conclusion that the μ₃-oxo species constitutes a critical structural element during catalysis (Supplementary Fig. 7).

### Functional significance of Asp14 in μ₃-hydroxide deprotonation in ShPPase

The mechanism by which the μ₃-hydroxide species is deprotonated in ShPPase has remained unresolved. Although recent structural studies have accumulated detailed insights into Family II PPases, the process by which deprotonation occurs within the trinuclear metal center has not been fully elucidated. Conventional models, based on Family I PPases, have proposed that nucleophilic attack proceeds via a μ-hydroxide possessing a free lone pair[18,29]. However, in the highly stable trinuclear metal configuration of Family II PPases, formation of such a lone pair would require partial dissociation of at least one metal ion, introducing a substantial energetic barrier.

To explore potential general base candidates responsible for deprotonating the μ₃-hydroxide, we examined the X-ray crystal structure of the PNP-bound form (PDB: 6LL8). Four aspartate residues were found in close proximity to the μ₃-hydroxide moiety (Fig. 1c, right). Among them, three residues (Asp12, Asp72, and Asp146) coordinate metal ions via both side-chain carboxylate oxygens, contributing to the formation and stabilization of the trinuclear center. In contrast, Asp14 engages only one of its carboxylate oxygens (Oδ1) in metal coordination, leaving the other (Oδ2) available for potential involvement in proton abstraction.

To test the functional role of Asp14, site-directed mutagenesis was performed. Both D14A and D14N variants exhibited drastic reductions in catalytic efficiency, with $k_{cat}/K_m$ values decreasing to approximately 1/4000–1/10,000 of that of the wild-type enzyme (Table 1). In contrast, the $K_m$ value of D14N is nearly identical to that of the wild type, whereas that of D14A is only approximately twofold higher (Table 1, Supplementary Fig. 8). This modest change in $K_m$ indicates that substrate binding and positioning are largely preserved and that any perturbation in substrate affinity is minor relative to the substantial loss of catalytic efficiency. In the trinuclear metal center, each metal ion coordinates one of the three terminal oxygen atoms of the substrate (Fig. 1b, c). Because substrate coordination is primarily mediated by the metal ions rather than directly by Asp14, the minimal change in $K_m$ suggests that the metal-assisted substrate-binding geometry remains essentially intact in the variants. These observations support preservation of the trinuclear metal architecture.

Consistently, DFT-optimized structures of the variants retain the trinuclear metal configuration (Supplementary Fig. 9). Together, the kinetic and computational results are consistent with preservation of the M1/M2/M3 metal sites under our assay conditions. Thus, the pronounced reduction

in activity is unlikely to arise from incomplete metal-cluster formation and instead reflects impairment of the chemical step—specifically, Asp14-mediated μ₃-hydroxide deprotonation required for nucleophile generation. Finally, sequence homology analysis revealed that Asp14 is strictly conserved across all Family II PPases (Supplementary Fig. 10), further supporting its essential role in μ₃-hydroxide deprotonation and catalytic nucleophile activation.

### S$_N$2-type hydrolysis catalyzed by a μ₃-oxo species in the trinuclear metal active site

To elucidate the catalytic mechanism of inorganic pyrophosphate (POP) hydrolysis in Family II PPase, we performed DFT calculations based on structural models incorporating POP as the substrate (Supplementary Figs. 5 and 6). The reaction proceeds through seven discrete states, including three transition states (TS) and two intermediates (IM), capturing the key structural and energetic transformations leading to product formation (Fig. 3). The mechanism unfolds in three sequential stages:

**Stage 1: hydrogen bond formation via rotation of Asp14.** In the crystal structures (PDB: 6LL7, 6LL8), Asp14 forms hydrogen bonds with Ser116 and Asn117. To investigate how Asp14 mediates deprotonation of the μ₃-hydroxide species, we constructed a rotation model of Asp14 (Supplementary Fig. 5). In the optimized reactant state, the Oδ1 atom of Asp14 coordinates a Zn²⁺ ion, while Oδ2 forms hydrogen bonds with the Oγ and N atoms of Ser116 at distances of 2.06 Å and 2.10 Å, respectively (Fig. 4). At this stage, the Oδ2–H(μ₃-hydroxide) distance is 3.70 Å.

Upon rotation of the Asp14 side chain, a hydrogen bond can form between Oδ2 and the μ₃-hydroxide, leading from the reactant state to the IM1 intermediate via TS1 (Fig. 4 and Supplementary Fig. 5). The imaginary-frequency vibrational mode of TS1 is shown in Supplementary Movie 1. Crystal structures indicate sufficient spatial allowance around Asp14 to accommodate this rotation. The transition from the reactant to TS1 requires only a small activation barrier of 2.9 kcal mol⁻¹ (ΔG), stabilizing the IM1 structure (Fig. 3). During this process, the original hydrogen bonds are disrupted, and a new hydrogen bond forms between Oδ2 and H(μ₃-hydroxide) at 1.43 Å (Fig. 4 and Supplementary Movie 1), priming the system for nucleophile activation.

**Stage 2: conversion of μ₃-hydroxide to μ₃-oxo.** This stage requires further rotation of the Asp14 side chain (Fig. 5 and Supplementary Fig. 6, Supplementary Movie 2). We defined a "rotation angle" of Asp14 (0° at IM1; positive away from the μ₃-oxo toward Asp72 Oδ2 and negative toward the μ₃-oxo) to quantify this movement.

In step 2-1, the Oδ2 atom of Asp14 approaches the proton of the μ₃-hydroxide, positioning it between the two oxygen atoms. As negative rotation proceeds (step 2-2), the proton is transferred to Oδ2, forming a covalent O–H bond and generating the μ₃-oxo species. The crossing of the Asp14 Oδ2–H and μ₃-oxo–H distances in this step reflects proton transfer (Fig. 5). Although this step raises the system's energy, the true transition state occurs later. Comparison of partial charges for the transferring proton in enzyme and non-enzyme models showed greater polarization by 0.436 in Mulliken charge in the enzymatic context, suggesting that the trinuclear metal center facilitates deprotonation by lowering the pKₐ of the hydroxide.

In step 2–3, the protonated Asp14 rotates further toward Asp72 via TS2 and forms a stabilizing hydrogen bond with its side chain (Fig. 5). The imaginary-frequency vibrational mode of TS2 is shown in Supplementary Movie 2. The rotation of protonated Asp14 corresponds to the rate-limiting step of the reaction. The point where the μ₃-oxo–H and Asp72 Oδ2–H distances cross coincides with the computed transition state, indicating that deprotonation completes as the hydrogen bond acceptor shifts from μ₃-oxo to Asp72.

**Stage 3: S$_N$2-type hydrolysis by the μ₃-oxo nucleophile.** In the final stage, the strongly nucleophilic μ₃-oxo species attacks the phosphorus atom (P1) of POP via TS3, forming the product complex with a low

**Fig. 4 | Hydrogen bond formation via side chain rotation of Asp14.** Upon rotation of the Asp14 side chain, a hydrogen bond forms between Oδ2 and the μ₃-hydroxide. Optimized structures of the reactant state (light green, upper left), the transition state TS1 (dark green, upper right), and the intermediate IM1 (white, lower right) are shown as sticks. Superpositions of Cα-aligned structures are boxed in gray. Only the active-site region is displayed to emphasize key residue rearrangements. Zn, Mg, O, N, P, and H atoms are colored slate purple, green, red, blue, orange, and white, respectively. For a complete model of Asp14 rotation, see Supplementary Fig. 5. The imaginary-frequency vibrational mode of TS1 is shown in Supplementary Movie 1.

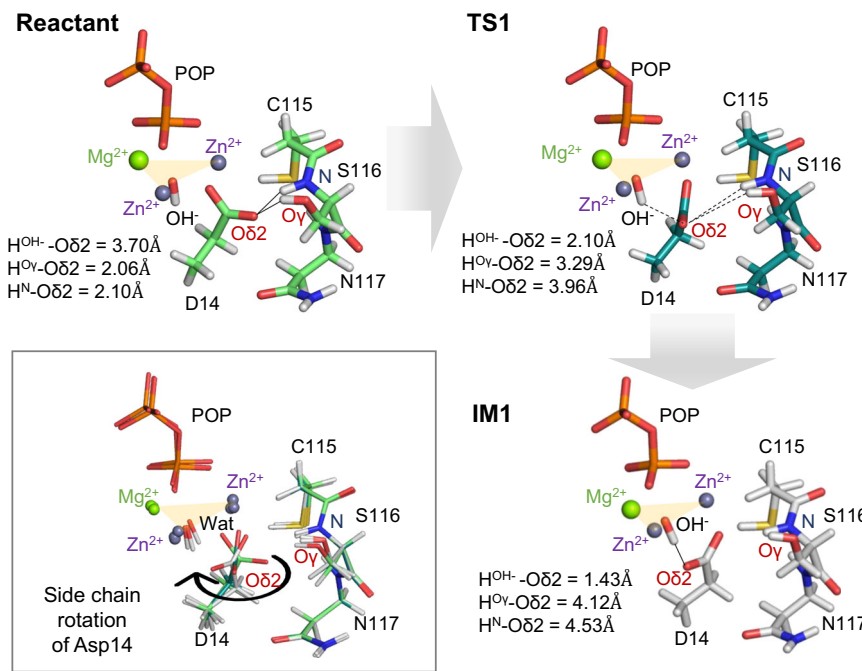

activation barrier of 2.2 kcal mol⁻¹ (Figs. 3, 6a). The imaginary-frequency vibrational mode of TS3 is shown in Supplementary Movie 3. In the IM2 state, the deprotonated μ₃-oxo is slightly offset from the Zn–Zn–Mg triangular plane and positioned 3.04 Å from P1 (Fig. 6a). At the TS, the μ₃-oxo aligns at the center of the trinuclear metal cluster, establishing an ideal geometry for in-line $S_N2$ attack. As the reaction proceeds from IM2 to the product state, the O–P1–μ₃-oxo angle shifts from 164.5° to 169.1°, approaching the optimal 180° alignment[30] supported by surrounding residues and metal ions.

DFT calculations delineated a plausible reaction pathway involving intermediates (IMs) and transition states (TSs) (Fig. 3). To clarify the reaction energetics, activation barriers are defined in terms of Gibbs free energies ($\Delta G$), with the corresponding electronic energies ($\Delta E$) provided in parentheses for comparison. The deprotonation of the μ₃-hydroxide proceeds via TS2 with an activation free energy ($\Delta G^{\ddagger}$) of 15.5 kcal mol⁻¹, defined as the free-energy difference between TS2 and the preceding intermediate IM1.

We also explored the possibility of a concerted mechanism in which proton abstraction and nucleophilic attack occur simultaneously. However, no single transition state combining both events could be located. Instead, geometry optimizations consistently converged to a stepwise pathway featuring a distinct intermediate (IM2) corresponding to the μ₃-oxo species prior to $S_N2$ attack. Frequency analysis confirms that IM2 is a true minimum on the free-energy surface, supporting a sequential mechanism.

Using the experimental $k_{cat}$ value of 99.8 s⁻¹ at 298 K (Table 1), the activation free energy estimated from the Eyring equation[31] is 14.7 kcal mol⁻¹, differing by only 0.8 kcal mol⁻¹ from the computed value. This level of agreement is well within the expected accuracy of DFT-based enzymatic reaction modeling. These results support the assignment of the IM1 → TS2 step as the rate-determining process and suggest that formation of the μ₃-oxo species enhances nucleophilic reactivity during phosphate bond cleavage.

Electrostatic potential (ESP) maps of IM1 and IM2 show the μ₃-oxo carrying a highly negative potential (red region), confirming its strong nucleophilic character (Supplementary Fig. 11). In contrast, the μ₃-hydroxide exhibits lower nucleophilicity (Supplementary Fig. 11). Mulliken charge analysis supports this, with the μ₃-oxo carrying a charge of –0.91 and the μ₃-hydroxide –0.83. These results provide a mechanistic rationale for the higher hydrolytic activity of Family II PPases relative to their Family I

counterparts, which lack the ability to generate a highly nucleophilic μ₃-oxo intermediate.

## Catalytic role of the trinuclear metal center in $S_N2$-type hydrolysis by Family II PPase

As demonstrated above, the μ₃-oxo species in Family II PPase functions as a highly nucleophilic intermediate and is central to catalytic activity. In this section, we further dissect the electronic and mechanistic properties of this intermediate using molecular orbital (MO) theory–based calculations. One of the principal functions of the trinuclear metal center is to lower the p$K$a of the μ₃-hydroxide oxygen, thereby facilitating its deprotonation and enabling the formation of a strongly nucleophilic oxo species. This structural arrangement not only promotes the generation of a reactive intermediate but also enhances the overall efficiency of the $S_N2$-type hydrolysis via multiple catalytic strategies.

MO analysis revealed that the electron density of the oxo ion (O²⁻) is more symmetrically distributed in the enzyme-catalyzed system compared to the non-enzymatic model, indicating stabilization of this inherently reactive species (Fig. 6b). Moreover, the trinuclear metal center positions the oxo nucleophile in an optimal geometry for attack on the phosphorus atom of POP, while simultaneously coordinating the three oxygen atoms of the substrate. In addition, the enzyme active site facilitates charge redistribution at the reaction center. Mulliken charge analysis showed that the phosphorus atom (P1) carries a charge of +1.522 in the enzymatic system versus +0.780 in the non-enzymatic system, indicating that coordination to the metal ions enhances the electrophilicity of the phosphorus by drawing electron density away.

During the transition from IM2 to TS3, the total charge of the three oxygen atoms bonded to P1 changes significantly in the non-enzymatic model (–0.146), but remains nearly constant in the enzymatic model (+0.006), clearly showing that metal–oxygen interactions stabilize the transition state.

These results collectively demonstrate that the trinuclear metal center of Family II PPase integrates multiple catalytic functions to facilitate highly efficient hydrolysis. In particular, deprotonation of the μ₃-hydroxide by Asp14 yields a highly reactive oxo species that is stabilized within the trinuclear environment. This arrangement ensures an ideal in-line configuration of the nucleophile and leaving group, thereby significantly lowering the activation energy compared to non-enzymatic systems.

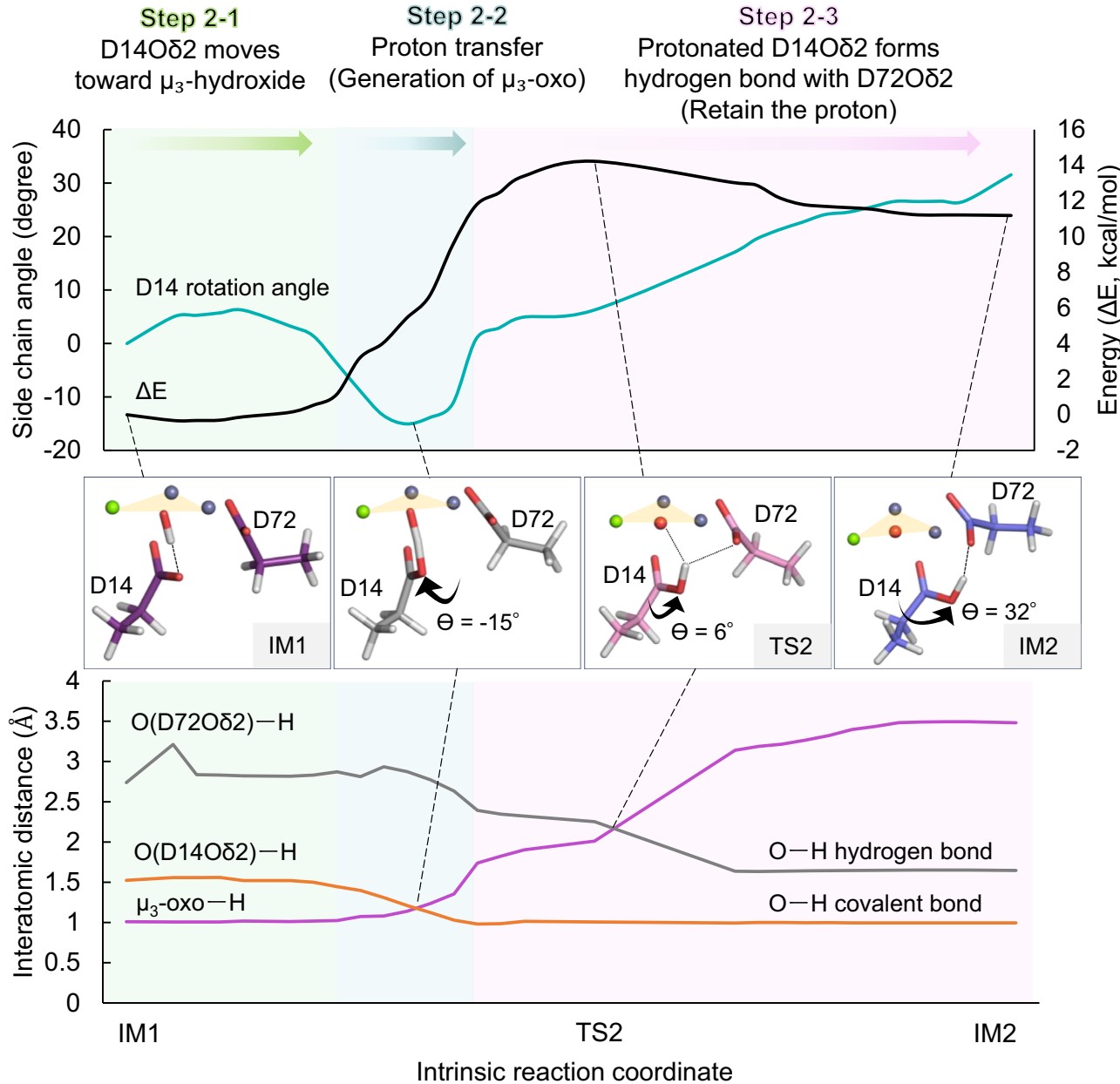

**Fig. 5 | Formation of the μ₃-oxo species.** Upper graph: rotation angle of Asp14 (light blue) and energy profile (black) during the transition from IM1 to IM2. Lower graph: interatomic distances: O(D72Oδ2)–H (gray), Asp14Oδ2–H (orange), and μ₃-oxo–H (pink). Three mechanistic steps—Step 2-1: Asp14Oδ2 approaches μ₃-OH (green), Step 2-2: proton transfer (blue), Step 2-3: protonated Asp14 forms H-bond with Asp72 (pink)—are highlighted. Optimized key intermediates and transition state structures are shown as bold stick models in black boxes. Atom coloring is as in Fig. 4. Full active-site models are shown in Supplementary Fig. 6. The imaginary-frequency vibrational mode of TS2 is shown in Supplementary Movie 2.

These insights provide a general mechanistic framework for metal-dependent cleavage of P–O bonds, particularly among enzymes classified as type E phosphatases. They also offer valuable guidance for the rational design of metal-based catalysts and artificial metalloenzymes. Furthermore, these findings may inform the development of metalloprotein-inspired therapeutics and biomimetic protein engineering strategies.

**Understanding the formation of active species**

Metalloproteins utilize a variety of highly evolved strategies to generate highly reactive transition metal species—such as metal–oxo/oxyl, metal–superoxo, and metal–(hydro)peroxo intermediates—that enable extremely efficient catalysis by significantly lowering activation energies[32]. However, the mechanisms underlying the formation of these active species remain poorly understood. For example, although a μ₃-oxo species has been proposed at the

trinuclear metal site of PhoX-type phosphatases involved in P–O bond cleavage, the specific formation pathway has yet to be elucidated[6,17].

In this study, we focused on the early activation step—i.e., the "induction phase"—to uncover the mechanism underlying the formation of the active nucleophile. Our results revealed that Asp14 acts as a general base, and that its unique conformational rotation plays a decisive role in facilitating deprotonation (Figs. 3–5). Structural analysis showed sufficient spatial flexibility around Asp14 to allow this rotation. In addition, site-directed mutagenesis of Asp14, combined with activity assays and sequence conservation analysis, confirmed its essential catalytic function (Table 1, Supplementary Fig. 8 and 10). These findings suggest that the precise spatial positioning of amino acid residues to act as catalytic bases represents a sophisticated and evolutionarily optimized mechanism for controlling active species formation in metalloproteins.

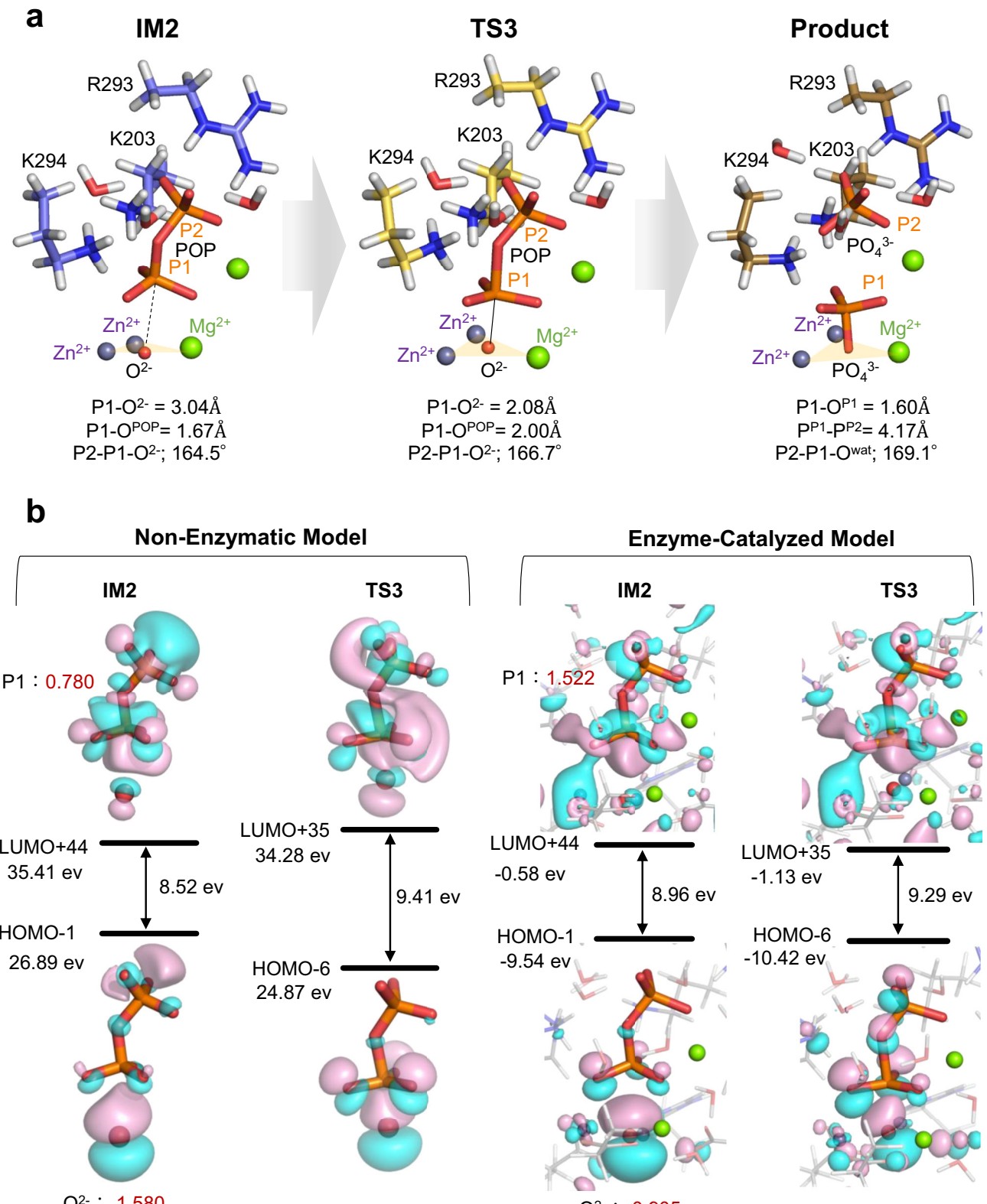

**Fig. 6 | $S_N2$ hydrolysis mechanism mediated by the $\mu_3$-oxo species. a** Optimized structures of IM2 (blue), TS3 (yellow), and product state (brown), with key interatomic distances and angles shown. **b** Molecular orbital maps comparing the non-enzymatic and enzyme-catalyzed models. Red numbers indicate Mulliken atomic charges. Atom coloring follows Fig. 4. Only the catalytic core is shown; for the full active site, see Supplementary Fig. 6. The imaginary-frequency vibrational mode of TS3 is shown in Supplementary Movie 3.

By integrating experimental XCS-EXAFS data with density functional theory (DFT) calculations, we have uncovered a complete mechanistic model for Family II PPase catalysis, including the crucial activation phase. Notably, we found that the energy barrier associated with $\mu_3$-oxo formation is higher than that of subsequent catalytic steps, indicating that the generation of the reactive intermediate constitutes the rate-limiting step of the entire reaction. This implies that active species formation, rather than substrate transformation per se, is the major determinant of reaction efficiency.

While many studies have focused on the hydrolysis pathway or product formation, the initial step of how the active species forms has often been overlooked. By highlighting this underappreciated yet fundamental step, our findings suggest new directions for enhancing catalytic performance and optimizing artificial enzymes and metal complexes. We propose that future efforts to understand and engineer metalloenzymes should place greater emphasis on optimizing the formation of active species.

## Conclusions

In this study, we elucidated the catalytic mechanism of Family II inorganic pyrophosphatase (PPase) using a combination of X-ray crystallography-based extended X-ray absorption fine structure (XCS-EXAFS) analysis, mutagenesis experiments, and density functional theory (DFT) calculations. We identified a key reaction pathway involving the deprotonation of a $\mu_3$-hydroxide intermediate by the conserved residue Asp14, followed by the formation of a highly nucleophilic $\mu_3$-oxo species that initiates $S_N2$-type hydrolysis of inorganic pyrophosphate. Quantum chemical analyses revealed that the trinuclear metal center facilitates oxo formation by lowering the p$K$a of the hydroxide species and stabilizing the reactive intermediate, aligns the $\mu_3$-oxo species for optimal nucleophilic attack, and stabilizes the transition state. These findings establish the formation of the $\mu_3$-oxo species as the rate-limiting step of catalysis and highlight the critical role of active species generation in determining reaction efficiency. Our study provides mechanistic insights into metal-dependent phosphate hydrolysis and offers a framework for the rational design of artificial metalloenzymes and bioinspired catalysts.

## Methods

### Materials and Protein Purifications

The recombinant wild-type ShPPase and ShPPase variants were expressed in *E. coli* BL21 (DE3) using pET16b as an expression vector and purified as previously described[33]. Metal-free ShPPase was prepared by EDTA treatment. The enzyme was diluted with 100 mM Tris/HCl (pH 7.5) buffer containing 20 μM EDTA and 50 mM KCl, subjected to three dilution/concentration cycles using ultrafiltration. The primer sequences were as follows: for the D14N mutation, 5′-TGATTCAGCTTCAATTTGTGGTGCAATCGC-3′(forward) and 5′-ATTGAAGCTGAATCAGGGATCTTGTGACCC-3′(reverse); and for the D14A mutation, 5′-TGATTCAAATTCAATTTGTGGTGCAATCGC-3′(forward) and 5′-ATTGAATTTGAATCAGGGATCTTGTGACCC-3′(reverse).

### Sample preparation for EXAFS

For EXAFS studies, 0.5 mg/mL metal-free ShPPase samples were incubated in an activation buffer containing 100 mM Tris/HCl (pH 7.5), 20 μM EDTA, 50 mM KCl, and 0.5 mM ZnCl$_2$ for two hours at 4 °C. Samples were desalted using spin columns two or three times to eliminate excess metals, which cause noise in EXAFS studies. During the spin column process, the metal-free buffer with 40 μM ZnCl$_2$ maintains two combined metals in the active site. This low concentration effect of ZnCl$_2$ can be ignored for EXAFS studies because it accounts for only 0.0004% of the total sample. Substrate-free Zn$^{2+}$-ShPPase sample was loaded into polypropylene PCR tubes, flash-frozen in liquid-nitrogen, and stored in a −80 °C freezer. PNP-bound Zn$^{2+}$-ShPPase sample contained final concentrations of 1.1 mM Zn$^{2+}$-ShPPase, 0.1 M PNP, and 0.1 M MgCl$_2$ at pH 7.5, and was stored in the same way as the substrate-free Zn$^{2+}$-ShPPase sample. Zn$^{2+}$-ShPPase samples had sufficient activity without extra metals (Supplementary Fig. 8, Supplementary Table 1).

### X-ray absorption spectroscopy

X-ray absorption fine structure (XAFS) is a structural technique that probes the local environment of a metal ion at a 0.01 Å-level resolution. For XAFS data collected at the metal $K$-edge, the edge region of the spectrum is sensitive to the effective charge on the metal ion and the coordination geometry, while the EXAFS region provides average metal–ligand bond distances and coordination numbers.

X-ray absorption measurements of each sample were obtained at the beamline 15 of the SAGA Light Source (SAGA-LS), with the ring operating at 1.4 GeV, 100–300 mA. A Si (111) double-crystal monochromator was used for energy selection at the Zn K-edge and data were measured in fluorescence mode as Zn Kα fluorescence using a 7-element Silicon drift detector (SDD, Techno-AP). The position of the SDD was adjusted to suppress the dead time to under 5%. All Zn XAFS spectra were collected by scanning the incident X-ray energy from 9.3310 to 10.2096 keV. The averaged data included 10 scans at about 42 minutes per scan for each sample to increase the signal-to-noise ratio. The sample was maintained under the supercooled environment in a stream of 100 K-nitrogen gas during data collection to minimize radiation damage. No photodegradation was observed for any of the samples from XANES data and activity assay (Supplementary Fig. 8 and Supplementary Table 2). In all experiments, individual scans were normalized to the incident photon flux and averaged using the program Athena from the software package Demeter[23]. For the averaged data, further processing of spectra including background subtraction and normalization was also performed using Athena, following standard protocols for X-ray spectroscopy described below. The data were normalized to an edge jump of 1.0 between background and spline curves at 9660 eV.

EXAFS fitting was performed using the program Artemis, also of the software package Demeter[23]. Possible scattering paths for the EXAFS models were initially determined using FEFF 7.0 in combination with a recent high-resolution crystal structure (PDB ID: 6LL7,6LL8). Each fit used a common value $\Delta E_0$ for every component in the fit. With a value of $S_0^2$, the scale factor was restricted to 0.8–1.0 for reproducibility. Experimental EXAFS data were converted to $k$ space by applying the EXAFS equation and subsequently weighted by $k^3$ to compensate for the damping of oscillations at high $k$. The $k^3$ data were fit over similar $k$-ranges ($k = 3$–$9$) using a nonlinear least-squares approach with theoretical values for Zn–O, Zn–N, Zn–P, and Zn–Zn bonds from FEFF 7.0 using structural refinement data for Zn$^{2+}$-ShPPase. EXAFS spectra were also Fourier transformed to produce radial structure functions (RSFs) that isolate frequency correlations between the central absorbing atom (Zn) and neighboring atoms as a function of bond distance ($R$). All data were fit in $R$-space using an $R$-range of 1 to 3.5 Å. Due to the complexity of the EXAFS of the ShPPase, fitting was limited to include only single-scattering paths. No smoothing was used at any point in any of the data processing.

### Activity assay and kinetic analysis

The activity was measured by the molybdenum blue method. A reaction mixture containing 10 μL of enzyme and 110 μL of 1 mM potassium pyrophosphate (K$_4$POP) as substrate in 100 mM Tris-HCl buffer, 50 mM KCl (pH 7.5), containing 5 mM MgCl$_2$, was incubated for 3 min at 25 °C. The reaction was stopped by the addition of 30 μL of 50 mM H$_2$SO$_4$. The reaction mixture was colored by the addition of 150 μL of 1% ammonium molybdate in 0.05% K$_2$SO$_4$ and 1% sodium ascorbate in Milli-Q water. The amount of phosphate liberated from the hydrolysis of inorganic POP was measured at 750 nm using a microplate reader (Thermo Fisher Scientific, Waltham, MA, USA) and a standard phosphate curve (0–500 μM phosphate) after 20 minutes. Specific activity (U/mg) was defined as 1 μmol of POP hydrolyzed per min per mg of protein. One unit of activity corresponded to the formation of 2 μmol of phosphate per min from 1 μmol of POP under the assay conditions. $K_m$ and $k_{cat}$ values were determined from velocity data at various concentrations of substrate using the program GraphPad Prism (GraphPad Software Inc.). The activation free energy

($\Delta G^{\ddagger}$) was estimated from $k_{cat}$ using the Eyring equation[31],

$$\Delta G^{\ddagger} = RT\ln(k_B T / h\, k_{cat})$$

where $k_B$ is the Boltzmann constant, $h$ is Planck's constant, $R$ is the gas constant, and $T$ is the absolute temperature.

## DFT calculations

First, the X-ray crystal structure (XCS) model was constructed based on the $Mg^{2+}$-ShPPase–PNP complex (PDB ID: 6LL8). Building upon this structural model, two additional computational models were constructed: an extended X-ray absorption fine structure (EXAFS) simulation model and a proposed hydrolysis reaction mechanism model (POP). The XCS model included the PNP substrate analogue, four Mg atoms, the F atom as a center of tri-metal structure, 11 water molecules, and the side chains of His8, Asp12, Asp14, Asp72, Asp93, His94, His95, Asp146, Lys203, Arg293, and Lys294. The total number of atoms and the total charge are 187 and +1, respectively. In the EXAFS model, two Mg atoms as binuclear metals were replaced by two Zn atoms. The F atom as a center of tri-metal structure was replaced by the O atom or OH molecule. The total number of atoms and the total charge are 188 and +1, respectively. In the POP model, the N atom of PNP was replaced with the O atom, the F atom as a center of tri-metal structure was replaced by the O atom or OH molecule, and two Mg atoms as binuclear metals were replaced by two Zn atoms. The total number of atoms and the total charge are 188 and +2, respectively. We also constructed the Asp14 rotation model for Asp14 rotation hypothesis, which added three side chains of Cys115, Ser116, and Asn117 to the POP model. The total number of atoms and the total charge are 222 and +2, respectively. In all calculated models, most of the side chains of amino acid residues were truncated at their $\beta$-carbon, which were converted to methyl groups. During geometry optimization, the $\beta$-carbon atoms were fixed to their crystallographic positions. This method is called the quantum chemical cluster approach, developed by Siegbahn, Himo, and de Visser, and is commonly used in enzyme reaction analysis[34,35]. For Lys203, Arg293, and Lys294 in each model, the frozen atom was selected to be the $\gamma$-carbon to limit the size of the Quantum Mechanics (QM) region to mimic the enzyme environment and prevent the unrealistic distortion of the cluster and ensure sufficient flexibility. The QM subsystem was treated using density function theory (DFT) calculations. All DFT calculation was performed using the B3LYP functional[36,37] and the 6–31 G(d) basis set[38,39] for H, C, N, O, P, S, F, and 6-311 + G(d) basis set[40,41] for Mg, and the Wachters basis set[42] with f polarization[43] for Zn, implemented with the Gaussian 16 program package[44]. The protonated model was characterized by the Amber force field (Amber10:EHT)[45] using the Molecular Operating Environment (MOE)[45]. Vibrational frequency analyses were performed at the same level of theory to confirm that the optimized structures correspond to true minima (no imaginary frequencies) or transition states (one imaginary frequency). However, the imaginary frequencies due to the constrained carbon atoms were ignored. All thermodynamic parameters and Gibbs energy corrections were obtained from these frequency calculations at 298.15 K. The reaction pathway was traced by the quasi-intrinsic reaction coordinate (quasi-IRC) method, followed to obtain the minimum-energy path, in which the TS structure was slightly perturbed in the direction of the reactants and products, followed by full geometry optimizations except the constrained carbon atoms. All molecular visualizations and animations of the imaginary-frequency vibrational modes of transition states were generated using Chemcraft version 1.8. The atomic coordinates for the optimized computational models are provided as a Supplementary Data 1 file.

## Reporting summary

Further information on research design is available in the Nature Portfolio Reporting Summary linked to this article.

## Data availability

The data that support the findings of this study are available within the paper and its Supplementary Information files. The atomic coordinates for the optimized computational models are provided as Supplementary Data 1. Source data underlying the figures and tables are provided as Supplementary Data 2. The protein crystal structures analyzed in this study are available from the Protein Data Bank under accession codes 6LL7 and 6LL8. All other raw data, including X-ray absorption spectroscopy and enzyme activity assay data, are available from the corresponding author upon reasonable request.

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

## Acknowledgements

This work was supported by a Grant-in-Aid for Scientific Research from the Ministry of Education, Culture, Sports, Science and Technology, Japan (JSPS KAKENHI; grant nos. JP23KJ1741 (S.M.), JP23K26836 (M.H.), JP24K21245 (K.Y.), and JP25K08678 (Y.S.)). We thank the Center for Advanced Instrumental and Educational Support of the Faculty of Agriculture (Kyushu University) for use of facilities for the enzyme assay.

## Author contributions

S.M. and K.W. conceived the study and designed the research. S.M. and M.H. prepared the protein samples. S.M., E.M., H.S., M.K., M.H., and K.W. performed the XAFS experiments, and S.M., E.M., H.S., and K.W. analysed the data. S.M., T.T., and Y.Kak. conducted the mutagenesis experiments. S.M., Y.S., K.Y., and K.W. designed the DFT calculations, S.M. and Y.Kam. carried out the calculations, and S.M., Y.Kam., Y.S., K.Y., T.T., Y.Kak., and K.W. discussed the results. S.M. and K.W. wrote the paper with input from all authors.

## Competing interests

The authors declare no competing interests.
