## [Transparent Peer Review file · Communications Chemistry]

μ_3 -Oxo nucleophile formation enables efficient S_N2 hydrolysis at the trinuclear metal center in inorganic pyrophosphatase

Corresponding Author: Professor Keiichi Watanabe

Version 0:

Reviewer comments:

Reviewer #1

(Remarks to the Author)

In this study, the authors define the catalytic mechanism of Family II inorganic pyrophosphatase (PPase) using XCS-EXAFS analysis, mutagenesis, and DFT calculations. The authors find that Asp14 deprotonates a μ_3 -hydroxide intermediate to generate a highly nucleophilic μ_3 -oxo species that performs an S_N2 -type attack on inorganic pyrophosphate. Quantum chemical analyses show that the trinuclear metal center effectively lowers the hydroxide pKa, stabilizes the μ_3 -oxo intermediate, positions it optimally for nucleophilic attack, and further electrostatically stabilizes the transition state. These results identify the μ_3 -oxo formation as the overall rate-limiting step and highlight the importance of reactive species generation in catalytic efficiency.

Overall, the study is carefully designed and executed. A few minor comments are below.

1.) p1, 2nd paragraph, the authors state: A prominent example is the cleavage of phosphoester (P–O) bonds involved in phosphate metabolism, particularly in pathways utilizing inorganic pyrophosphate (POP) and its derivatives^{4,5}.

Please note that in pyrophosphate this is a phosphoanhydride bond. This reviewer suggest to mention both phosphoester and phosphoanhydride bonds in this regard.

2.) The authors assign Asp14 as the key acid-base catalyst for deprotonation of the hydroxide. In line with this hypothesis, the corresponding variants (D14N, D14A) show only very little enzymatic activity. The authors argue that the tri-metal center is - based on calculations - still formed in the variants. Could they test this experimentally with EXAFS analysis of at least one variant? D14 binds with one of its oxygens to one metal ion and it appears important to rule out that the loss of activity results from incomplete metal cluster formation rather than eliminated acid-base chemistry.

Reviewer #2

(Remarks to the Author)

This study provides a clear and well-supported mechanistic explanation for how Family II inorganic pyrophosphatases (PPases) achieve efficient P–O bond hydrolysis through a trinuclear metal centre. The authors demonstrate that a μ_3 -oxo species (not a μ_3 -hydroxide) is the active nucleophile responsible for in-line S_N2 attack, and they show that residue Asp14 functions as the general base. Its rotation initiates μ_3 -oxo formation and contributes to the rate-limiting step. The authors integrated spectroscopic (XCS-EXAFS) data, site-directed mutagenesis and DFT calculations to develop a coherent mechanistic model. EXAFS provided key solution-state evidence for substrate-induced structural changes, while the computational data demonstrate how the trinuclear metal cluster lowers the hydroxide pKa, stabilizes the μ_3 -oxo intermediate, and enhances phosphorus electrophilicity. A few sections in the study, however, require some clarifications or extra work, particularly the interpretation of EXAFS data from heterogeneous metal samples and the use of a mixed Zn/Zn/Mg cluster in DFT modelling. Addressing these points will enhance the impact of the structural and energetic conclusions. Overall, the study offers strong mechanistic insight and merits publication following some relatively minor revisions.

Suggested revisions/improvements:

- 1) Add an explanation of how the Zn–O(μ_3 -oxo) distance was reliably extracted despite the significant proportion of uninuclear or inactive species in the Zn²⁺ EXAFS samples. This would add some clarity to how the fitting procedure differentiates the active-site signal.
- 2) Define the activation barrier corresponding to the rate-limiting step. Does the 15.5 kcal/mol barrier correspond to the free-energy difference between the reactant state (R) and the highest transition state (TS2)?
- 3) Can you elaborate on the relatively modest K_m increase in the Asp14 mutants? The two-fold increase in K_m for D14A and D14N may indicate either a minor substrate-positioning effect or is negligible relative to the much larger loss in catalytic efficiency.
- 4) The visualisation of the DFT cluster models could be improved. For instance, an additional supplementary figure isolating Asp14, Ser116, Asn117 and the μ_3 -hydroxide/oxo species within the larger QM cluster could make it easier to follow proposed conformational changes driving the μ_3 -oxo formation.

Reviewer #3

(Remarks to the Author)

In this manuscript, Watanabe et al. present a clear study on the SN2-type hydrolysis catalysed by a tri-nuclear metal-center pyrophosphatase. The authors have investigated the reaction mechanism and proposed a plausible catalytic pathway by combining experimental data with DFT calculations. Overall, the study is carefully conducted and of interest to the field. I have a few comments and suggestions that may help improve the clarity and completeness of the manuscript.

1. To investigate the catalytic mechanism of the inorganic pyrophosphatase, the authors employed DFT calculations. During geometry optimizations using the Gaussian 16 program, several β -carbon atoms of selected amino acid residues in the cluster models were fixed. This is a commonly used and reasonable strategy in enzymatic mechanism studies. However, the authors did not cite relevant references for this methodology, namely the quantum chemical cluster approach developed by Siegbahn, Himo, and de Visser, among others. It would be appropriate to cite representative references to support the use of this approach.

2. The 6-311+G(d) basis set is generally suitable for magnesium; however, it may not be good for zinc. It would be advisable to use a more appropriate basis set for Zn to evaluate the reaction energy profile better.

3. Figure 6 presents the calculated energy profile of the SN2 mechanism. According to the figure, the overall energy barrier appears to be 16.7 kcal/mol (=10.3 + 6.4 kcal/mol). However, in the main text (page 8), an overall barrier of 15.5 kcal/mol is reported. This discrepancy should be clarified. In addition, it would be informative to compare the calculated energy barrier with experimentally measured reaction rates, if available.

4. The proposed mechanism proceeds in a stepwise manner: Asp14 first abstracts a proton from the bridging hydroxide, followed by nucleophilic attack of the resulting oxo anion on the phosphorus center. It would be interesting to discuss whether these two steps could occur in a concerted fashion. In such a scenario, a single transition state might exist after the rotation of Asp14.

Other suggestions:

1. For clarity, it would be helpful to provide captions for all figures in the manuscript.
2. On page 12, the PDB ID should be written as "6LL8" rather than "6ll8."

Reviewer #1:**Comment 1:**

p1, 2nd paragraph, the authors state: A prominent example is the cleavage of phosphoester (P–O) bonds involved in phosphate metabolism, particularly in pathways utilizing inorganic pyrophosphate (POP) and its derivatives^{4,5}.

Response:

Thank you for this helpful comment. As suggested, we have revised the text to refer to both phosphoanhydride and phosphoester bonds. (line 50)

Comment 2:

The authors assign Asp14 as the key acid-base catalyst for deprotonation of the hydroxide. In line with this hypothesis, the corresponding variants (D14N, D14A) show only very little enzymatic activity. The authors argue that the tri-metal center is - based on calculations - still formed in the variants. Could they test this experimentally with EXAFS analysis of at least one variant? D14 binds with one of its oxygens to one metal ion and it appears important to rule out that the loss of activity results from incomplete metal cluster formation rather than eliminated acid-base chemistry.

Response:

We thank the reviewer for this important and insightful suggestion. EXAFS measurements require advance reservation of synchrotron beamtime, and additional experiments cannot be completed within the revision timeline.

Nevertheless, several lines of evidence argue against disruption of the trinuclear metal cluster in the Asp14 variants. As also noted by Reviewer #2 (Comment 3), the K_m value of the D14N variant is nearly identical to that of the wild type, and that of D14A is only approximately twofold higher. This modest change in K_m indicates that substrate binding and positioning are largely preserved and that any perturbation in substrate affinity is minor relative to the substantial reduction in catalytic efficiency.

In the trinuclear metal center, each metal ion coordinates one of the three terminal oxygen atoms of the substrate (Figs. 1b and 1c). Because substrate coordination is primarily mediated by the metal ions rather than directly by Asp14, the minimal change in K_m suggests that the metal-assisted substrate-binding geometry remains essentially intact in the variants. This observation is inconsistent with major disruption of the metal cluster.

Furthermore, DFT-optimized structures of the variants retain the trinuclear metal configuration, supporting preservation of the M1/M2/M3 metal sites. Taken together, the kinetic and computational results are consistent with structural integrity of the trinuclear metal center under our assay conditions. We therefore conclude that the pronounced reduction in activity is unlikely to arise from incomplete metal-cluster formation and instead reflects impairment of the acid–base chemistry mediated by Asp14. This interpretation has been clarified in the revised manuscript (lines 195–210).

Reviewer #2:

Comment 1:

Add an explanation of how the Zn–O(μ_3 -oxo) distance was reliably extracted despite the significant proportion of uninuclear or inactive species in the Zn²⁺ EXAFS samples. This would add some clarity to how the fitting procedure differentiates the active-site signal.

Response:

We thank the reviewer for this important comment regarding sample heterogeneity and the reliability of the EXAFS analysis. We acknowledge that the Zn²⁺-bound preparation contains a mixture of binuclear and uninuclear species.

To assess the robustness of the extracted Zn–O(μ_3 -oxo) distance, we performed systematic fitting tests, now summarized in Table S3 of the Supporting Information (lines 138–142). Models lacking a distinct short Zn–O shell resulted in significantly poorer fits (R-factor > 0.048) compared with the best-fit model (R = 0.035). In addition, constraining the Zn–O distance to 2.0 Å—representative of non-bridging water in uninuclear species—led to a clear deterioration in fit quality (R-factor increased from 0.035 to 0.047; χ^2 increased from 67.05 to 89.24). These results indicate that inclusion of the short \sim 1.82 Å Zn–O shell is statistically justified and necessary to adequately reproduce the experimental spectrum.

We have also clarified in the revised manuscript the ensemble-averaged nature of the EXAFS signal (lines 142–145). Because the uninuclear species are expected to exhibit longer Zn–O distances, their contribution would shift the averaged value toward longer bond lengths. Therefore, the observed 1.82 Å distance represents a conservative upper estimate for the active binuclear species, implying that the intrinsic Zn–O(μ_3 -oxo) distance in the catalytically competent site is likely equal to or shorter than this value.

Together, the statistical fitting results and the physical interpretation support the reliability of the extracted Zn–O(μ_3 -oxo) distance despite the presence of mixed species.

Comment 2:

Define the activation barrier corresponding to the rate-limiting step. Does the 15.5 kcal/mol barrier correspond to the free-energy difference between the reactant state (R) and the highest transition state (TS2)?

Response:

We thank the Reviewer for raising this important point regarding the definition of the activation barrier.

In the original version of the manuscript, the reaction profile was presented primarily in terms of electronic energies (ΔE), which may have caused ambiguity in the definition of the activation barrier. In the revised manuscript, Fig. 3 (formerly Fig. 6) has been updated to display Gibbs free energies (ΔG) as the primary values, with the corresponding ΔE values provided in parentheses. This ensures that the activation barrier is consistently defined in terms of free energy, in accordance with transition-state theory.

The activation barrier for the rate-limiting step is defined as the free-energy difference between a given intermediate and the subsequent transition state. In the present profile, TS2 is the highest transition state and is preceded by IM1 (–2.4 kcal/mol relative to the reactant). Accordingly,

$$\begin{aligned}\Delta G^\ddagger &= G(\text{TS2}) - G(\text{IM1}) \\ &= 13.1 - (-2.4) \\ &= 15.5 \text{ kcal/mol.}\end{aligned}$$

Thus, the reported 15.5 kcal/mol barrier does not correspond to the free-energy difference between the reactant state (R) and TS2 (which is 13.1 kcal/mol), but rather to the elementary step IM1 \rightarrow TS2. This step is the highest barrier along the reaction coordinate and therefore determines the overall rate.

The manuscript has been revised accordingly to clarify both the energetic representation and the definition of the activation barrier (Fig. 3, lines 264–269). We appreciate the Reviewer's comment, which has improved the clarity of our presentation.

Comment 3:

Can you elaborate on the relatively modest K_m increase in the Asp14 mutants? The two-fold increase in K_m for D14A and D14N may indicate either a minor substrate-positioning effect or is negligible relative to the much larger loss in catalytic efficiency.

Response:

We thank the reviewer for this insightful comment.

The approximately twofold increase in K_m observed for the D14A and D14N variants is modest compared to the several orders of magnitude reduction in catalytic efficiency (k_{cat} and k_{cat} / K_m). While this small change may reflect a minor perturbation in substrate positioning, it is negligible relative to the pronounced loss of catalytic activity and indicates that overall substrate binding is largely preserved.

Structurally, in the trinuclear metal center, each metal ion coordinates one of the three terminal oxygen atoms of the substrate (Figs. 1b and 1c). Because substrate coordination is primarily mediated by the metal ions rather than directly by Asp14, substitution of Asp14 is not expected to substantially disrupt substrate-binding geometry. The near-wild-type K_m of D14N and the limited twofold increase in D14A therefore suggest that the mutations exert, at most, a minor effect on substrate positioning.

In contrast, the dramatic decrease in catalytic efficiency supports the conclusion that Asp14 plays a critical role in acid–base chemistry, specifically in promoting μ_3 -hydroxide deprotonation to generate the reactive μ_3 -oxo nucleophile. Thus, the kinetic data are most consistent with a mechanism in which Asp14 primarily contributes to reactive species generation rather than substrate binding. This interpretation has been clarified in the revised manuscript (lines 195–210).

Comment 4:

The visualisation of the DFT cluster models could be improved. For instance, an additional supplementary figure isolating Asp14, Ser116, Asn117 and the μ_3 -hydroxide/oxo species within the larger QM cluster could make it easier to follow proposed conformational changes driving the μ_3 -oxo formation.

Response:

We thank the Reviewer for this constructive suggestion.

Figure 4 already highlights Asp14, Ser116, Asn117, the μ_3 -hydroxide/oxo species, and the metal centers in isolation from the surrounding protein environment, allowing their interactions and orientations to be clearly visualized.

However, we agree that static representations (Figs. 4 and 5) may not fully capture the continuous conformational changes associated with Asp14 rotation and subsequent μ_3 -hydroxide deprotonation leading to μ_3 -oxo formation.

To better illustrate these dynamic processes, we have added MP4 animations for all transition states (Movies S1–S3) to the Supporting Information. These animations depict the vibrational modes corresponding to the imaginary frequencies of each transition state:

Movie S1 (TS1): Formation of the hydrogen-bonding interaction between Asp14 and the μ_3 -hydroxide via Asp14 rotation.

Movie S2 (TS2): Proton transfer and reorientation of protonated Asp14 toward Asp72 (rate-limiting step).

Movie S3 (TS3): SN2-type hydrolysis mediated by the μ_3 -oxo nucleophile.

These dynamic visualizations provide a clearer representation of the conformational and proton-transfer events underlying μ_3 -oxo formation.

Reviewer #3:

Comment 1:

To investigate the catalytic mechanism of the inorganic pyrophosphatase, the authors employed DFT calculations. During geometry optimizations using the Gaussian 16 program, several β -carbon atoms of selected amino acid residues in the cluster models were fixed. This is a commonly used and reasonable strategy in enzymatic mechanism studies. However, the authors did not cite relevant references for this methodology, namely the quantum chemical cluster approach developed by Siegbahn, Himo, and de Visser, among others. It would be appropriate to cite representative references to support the use of this approach.

Response:

As pointed out by the reviewers, we have cited the following two papers and revised the computational methods section.

Himo, F. & de Visser, S. P. Status Report on the Quantum Chemical Cluster Approach for

Modeling Enzyme Reactions. *Commun. Chem.* **5**, Article number: 29 (2022).

Siegbahn, P. E. M. & Himo, F. The Quantum Chemical Cluster Approach for Modeling Enzyme Reactions. *WIREs Comput. Mol. Sci.* **1**, 323–336 (2011)

We added the sentence “This method is called the quantum chemical cluster approach, developed by Siegbahn, Himo, and de Visser, and is commonly used in enzyme reaction analysis.” to the calculation section (lines 468–470).

Comment 2:

The 6-311+G(d) basis set is generally suitable for magnesium; however, it may not be good for zinc. It would be advisable to use a more appropriate basis set for Zn to evaluate the reaction energy profile better.

Response:

The 6-311+G(d) basis set for zinc uses the Wachters basis with f polarization corresponding to a (14s9p5d1f/9s5p3d1f) type contraction, making it a higher precision basis function compared to the standard 6-311G(d) basis. Due to insufficient literature citations, the following references were added (lines 475–476).

Wachters, A. J. H. Gaussian basis set for molecular wavefunctions containing third-row atoms. *J. Chem. Phys.* **52**, 1033–1036 (1970).

Raghavachari, K. & Trucks G. W. Highly correlated systems: Excitation energies of first row transition metals Sc-Cu. *J. Chem. Phys.* **91** 1062-1065 (1989).

Comment 3:

Figure 6 presents the calculated energy profile of the SN2 mechanism. According to the figure, the overall energy barrier appears to be 16.7 kcal/mol (=10.3 + 6.4 kcal/mol). However, in the main text (page 8), an overall barrier of 15.5 kcal/mol is reported. This discrepancy should be clarified. In addition, it would be informative to compare the calculated energy barrier with experimentally measured reaction rates, if available.

Response:

We thank the Reviewer for carefully examining the calculated energy profile and for highlighting this apparent discrepancy.

In the original version of the manuscript, the energy diagram was presented primarily in terms of electronic energies (ΔE), which may have created ambiguity in how the overall barrier was interpreted. In particular, the value of 16.7 kcal/mol arises from summing two electronic energy differences (10.3 + 6.4 kcal/mol), which does not correspond to the definition of the activation barrier within transition-state theory.

In the revised manuscript, Fig. 3 (formerly Fig. 6) has been updated to display Gibbs free energies (ΔG) as the primary values, with ΔE values shown in parentheses. The activation barrier is defined as the free-energy difference between a given intermediate and the subsequent transition state. In the present reaction profile, TS2 is the highest transition state and is preceded by IM1 (-2.4 kcal/mol relative to the reactant). Accordingly,

$$\begin{aligned}\Delta G^\ddagger &= G(\text{TS2}) - G(\text{IM1}) \\ &= 13.1 - (-2.4) \\ &= 15.5 \text{ kcal/mol.}\end{aligned}$$

Thus, the overall barrier is 15.5 kcal/mol and corresponds to the elementary step IM1 \rightarrow TS2, which represents the highest point along the free-energy surface and therefore determines the overall rate.

Regarding comparison with experiment, using the reported k_{cat} value of 99.8 s⁻¹ at 298 K, the activation free energy was estimated via the Eyring equation:

$$\Delta G^\ddagger = RT \ln (k_B T / h k_{\text{cat}}).$$

This yields an experimental activation free energy of approximately 14.7 kcal/mol. The calculated barrier (15.5 kcal/mol) therefore differs by only ~0.8 kcal/mol from experiment, which is well within the expected accuracy of DFT-based enzymatic reaction modeling. This close agreement further supports the validity of the proposed mechanism.

The manuscript has been revised accordingly to clarify the definition of the activation barrier and to include this quantitative comparison (lines 264–281, 441–447).

We appreciate the Reviewer's comment, which helped improve the clarity and consistency of the energetic analysis.

Comment 4:

The proposed mechanism proceeds in a stepwise manner: Asp14 first abstracts a proton from the bridging hydroxide, followed by nucleophilic attack of the resulting oxo anion on the phosphorus center. It would be interesting to discuss whether these two steps could occur in a concerted fashion. In such a scenario, a single transition state might exist after the rotation of Asp14.

Response:

We thank the Reviewer for this insightful suggestion regarding the possibility of a concerted mechanism.

In the revised manuscript, the reaction profile is presented in terms of Gibbs free energies (ΔG). As shown in Fig. 3, the highest transition state along the reaction coordinate is TS2, corresponding to Asp14-mediated deprotonation of the μ_3 -hydroxide, with an activation free energy of 15.5 kcal mol⁻¹ relative to IM1. The subsequent nucleophilic substitution step proceeds via TS3, whose free energy is significantly lower than that of TS2.

Importantly, we explicitly explored the possibility of a concerted pathway in which proton abstraction and nucleophilic attack occur simultaneously following rotation of Asp14. Despite systematic searches using multiple initial guess structures, no single transition state combining both events could be located. Instead, all optimizations converged to a stepwise pathway featuring a distinct intermediate (IM2) corresponding to the μ_3 -oxo species prior to S_N2 attack. Frequency analysis confirms that IM2 is a true minimum on the free-energy surface.

These results indicate that the reaction proceeds through a sequential mechanism in which deprotonation precedes nucleophilic attack, rather than via a concerted transition state. We have clarified this point in the revised manuscript (lines 270–275).

Other suggestions:

1. For clarity, it would be helpful to provide captions for all figures in the manuscript.
2. On page 12, the PDB ID should be written as “6LL8” rather than “6ll8.”

Response:

3. We have provided captions for all figures in the revised manuscript.
4. The PDB ID has been corrected from “6ll8” to “6LL8”.